# Aligning Self-Interested Agents with Welfare Maximization in Non-cooperative Equilibrium via Mechanism Design

## Abstract

Self-interested multi-agent reinforcement learning has attracted growing attention for its applicability in real-world scenarios. In such settings, social dilemmas often arise, where agents prioritize individual gains over social welfare. Therefore, addressing these social dilemmas is critical for improving social welfare. However, prior work has notable limitations: (1) opponent modeling and incentive design approaches rely heavily on access to other agents' internal parameters and detailed information. As the number of agents increases or access to such information becomes limited, accurately modeling others' impact becomes difficult, leading to degraded performance; (2) centralized training is often ineffective, as relying on a single global training signal fails to capture the heterogeneous objectives and behaviors of self-interested agents, limiting effective individual policy learning. To overcome these limitations, we propose a mechanism design approach that leverages centralized information rather than centralized learning, without requiring access to other agents' internal parameters. Such mechanism dynamically reshapes each agent's reward to align individual incentives with social welfare. Building on this mechanism, we develop a value iteration algorithm that integrates a counterfactual critic and a maximized return predictor, further improving learning effectiveness. Extensive experiments on social dilemma environments demonstrate that our method achieves higher social welfare compared with existing baselines.

## 1 Introduction

Self-interested multi-agent systems, in which each agent optimizes its own cumulative reward while neglecting the impact on others, are prevalent across many real-world domains such as resource allocation (Parvini et al., 2023; Ye et al., 2019; Wang et al., 2025), transportation systems (Bnaya et al., 2013; Bonnefon et al., 2016), and market economies (Liu et al., 2022b). However, such self-interested behaviors often give rise to *social dilemmas* (Kollock, 1998), where conflicting goals drive agents to act recklessly in pursuit of their personal self-interest. The resulting social dilemmas typically yield social welfare significantly worse than the Pareto-optimal outcome (Leibo et al., 2017). This motivates the need for effective methods to mitigate conflicts of interest and improve social welfare in self-interested multi-agent settings.

Prior studies have explored *opponent modeling* (Foerster et al., 2017; Letcher et al., 2018; Zhou et al., 2024) and *incentive design* (Ratliff et al., 2019; Yuan et al., 2022), both of which rely on access to other agents' internal parameters or detailed behavioral information in order to accurately model their strategies and interactions (Yang et al., 2020; Guresti et al., 2023). When such access is available, these methods can yield promising results, as agents are able to better anticipate others' responses and adapt their own strategies accordingly. However, as the number of agents increases or when access to such information becomes restricted, modeling others' impact becomes increasingly inaccurate and computationally expensive, often leading to unstable learning and degraded social welfare. Another line of work based on *centralized training* (Foerster et al., 2018; Rashid et al., 2018), attempts to provide a unified global signal to coordinate agents toward a shared objective. Yet, in self-interested multi-agent reinforcement learning (MARL) settings where agents pursue inherently conflicting goals, a single centralized signal fails to capture individual objectives and heterogeneity, resulting in poor credit assignment and ineffective learning dynamics (Parvini et al., 2023; Paccagnan et al., 2020).

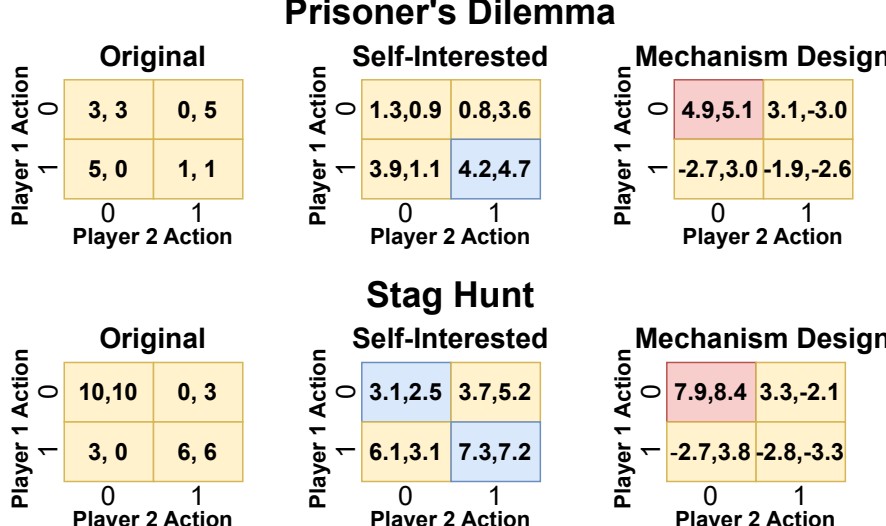

Figure 1: Didactic examples of two classical social dilemmas. The top row shows the Prisoner's Dilemma and the bottom row shows the Stag Hunt. In each case, the left matrix represents the original payoffs, the middle matrix shows the outcomes learned by self-interested agents, and the right matrix illustrates the reshaped payoffs generated by mechanism design. Here, action 0 denotes cooperate in Prisoner's Dilemma or stag in Stag Hunt, and action 1 denotes defect in Prisoner's Dilemma or hare in Stag Hunt.

To address these limitations, we propose the *Mechanism Design-based Markov Game (MDMG)* framework. Instead of requiring access to other agents' internal parameters or private rewards, MDMG introduces a centralized yet lightweight mechanism that leverages aggregated observations from all agents to reshape their incentives. This centralized module does not participate in policy optimization or control. Instead, it dynamically adjusts each agent's reward function such that maximizing individual returns becomes inherently aligned with improving collective social welfare. In this way, MDMG transforms the original game into one where self-interested learning naturally converges to welfare-enhancing outcomes, yielding a *Mechanism Design Induced Equilibrium (MDIE)*. Fig. 1 illustrates this idea using two classical social dilemmas—Prisoner's Dilemma and Stag Hunt. The left matrices show the original payoffs, where self-interested agents (middle) converge to Nash equilibria (blue cells) that yield lower welfare. In contrast, the right matrices depict the reshaped payoffs under our mechanism design, where both agents achieve higher individual payoffs and jointly reach socially optimal outcomes (red cells). Through this reshaping process, MDMG aligns self-interested behaviors with social welfare without altering the agents' decentralized optimization process. We summarize our contributions in this paper as follows:

- We formally define the MDMG framework to address the dilemmas in self-interested MARL, ensuring that self-interested optimization is naturally aligned with social welfare.

- Our method employs a centralized mechanism that reshapes agents' rewards using centralized information, without relying on others' internal parameters or a single global signal, ensuring effectiveness and robustness.

- We provide theoretical guarantees of the existence and convergence for the MDIE, and empirically validate the effectiveness of our method in escape room, multi-agent bidding, and bandwidth allocation in communication channels, where it significantly outperforms existing baselines.

## 2 Related Work

### 2.1 Self-Interested MARL

Self-interested MARL has been widely studied in domains such as social dilemmas and competitive resource allocation (Carmel & Markovitch, 1995; He et al., 2016; Foerster et al., 2017; Cavallo et al., 2012). One

major line of work focuses on *opponent modeling*, where agents estimate or predict the strategies of others to improve their own decision-making (Carmel & Markovitch, 1995; He et al., 2016; Liu et al., 2022a; Yu et al., 2022). For instance, Letcher et al. (2018) proposes a robust opponent modeling algorithm that avoids overly aggressive behaviors, while Zhou et al. (2024) introduces a reciprocal reward structure that accounts for the effect of one agent's actions on others. Another main direction is *incentive design*, where agents or external designers modify reward signals to align individual behaviors with other agents' impact. For example, Mguni et al. (2019) proposes learning methods to incentivize others, where each agent is equipped with an incentive designer that learns to modify the local reward functions of others through additional reward signals. In addition, Sen (1996) introduces probabilistic reciprocity incentives, where agents stochastically decide whether to cooperate based on the balance of past interactions and the relative cost of cooperating with others. Although effective in specific environments, both opponent modeling and incentive design generally require access to other agents' internal information or parameters in order to predict their behaviors or estimate their impact. When such access is restricted or the number of agents increases, estimation becomes unreliable and performance degrades.

## 2.2 Mechanism Design

An alternative line of research on self-interested multi-agent systems is grounded in *mechanism design*, a framework from economics and game theory that focuses on designing the rules of the game so that individually rational behaviors also lead to better outcomes (Li et al., 2024; Paccagnan et al., 2022; Crawford & Veloso, 2006; Xuan et al., 2024). In self-interested MARL, mechanism design often involves a centralized module that reshapes the rewards or redistributes credits among agents to provide more informative feedback. For example, Crawford & Veloso (2006) applies mechanism design as a central planner to solve auction problems, while Vulkan & Jennings (1998) adopts mechanism design to improve negotiation processes. Recent works further integrate mechanism design with neural networks (Shen et al., 2018; Wang et al., 2020), enabling more flexible and scalable approaches to real-world applications. Inspired by this perspective, we use mechanism design to reshape the reward structure in MARL, so that self-interested agents converge to equilibria consistent with social welfare.

## 3 Methodology

In this section, we first formalize the setting of **(1) Mechanism Design-based Markov Games**, where the objective is to achieve the Mechanism Design Induced Equilibrium. We then introduce **(2) the mechanism design**, which consists of a centralized module along with two key components: a counterfactual critic to evaluate the impact of individual actions on joint state-action value, and a maximized return predictor (MRP) to avoid convergence to the local optimum. Finally, we describe **(3) the value iteration algorithm**, including the update rules, convergence properties, and practical details that ensure agents can consistently align their self-interested policies with the mechanism design reward.

### 3.1 Mechanism Design-based Markov Games

We extend the classical Markov decision process (MDP) (Littman, 1994) to MDMG, where the self-interested agents achieve the best social welfare using the mechanism design reward $\bar{R}$. Formally, MDMG is defined by a tuple:

$$\mathcal{M} = \langle \mathcal{S}, \mathcal{A}, P, R, \bar{R}, N, \gamma \rangle, \tag{1}$$

where $\mathcal{S}$ represents the set of states describing the environment. $\mathcal{A} = \mathcal{A}_1 \times \mathcal{A}_2 \times \cdots \times \mathcal{A}_N$ denotes the set of joint actions, where $\mathcal{A}_i$ is the action space of agent $i$, and $N$ is the total number of agents. The $P : \mathcal{S} \times \mathcal{A} \times \mathcal{S} \to [0,1]$ defines the state transition function, specifying the probability of transitioning to state $s'$ given the current state $s$ and joint action $\boldsymbol{a} \in \mathcal{A}$. Each agent follows a joint policy $\pi = (\pi_1, \ldots, \pi_N)$, where $\pi : \mathcal{S} \to \mathcal{A}$. At each timestep, every agent $i$ selects an action according to its policy, $a_i \sim \pi_i(s)$. The joint action $\boldsymbol{a} = (a_1, a_2, \ldots, a_N)$ determines the state transition $s' \sim P(s' \mid s, \boldsymbol{a})$, where $s'$ is the new state. $R = \{R_1, R_2, \ldots, R_N\}$ represents the reward functions generated by the environment, where $R_i : \mathcal{S} \times \mathcal{A} \to \mathbb{R}$ is the reward function for agent $i$. $\gamma \in [0,1)$ is the discount factor, capturing the trade-off between immediate

and future rewards. Additionally, $\bar{R} = \{\bar{R}_1, \bar{R}_2, \ldots, \bar{R}_N\}$ is the mechanism design reward, where $\bar{R}_i$ is the reward for agent $i$ generated by the mechanism design.

In standard Markov games, the value function $V_i(s)$ for agent $i$ represents the expected cumulative reward $R_i$, obtained when starting from state $s^0$:

$$V_i(s) = \mathbb{E}\left[\sum_{t=0}^{\infty} \gamma^t R_i(s^t, \boldsymbol{a}^t) \mid s^0 = s, \boldsymbol{a}^t \sim \pi\right]. \tag{2}$$

In our setting, social welfare is defined as the sum of value functions for all agents under a joint policy $\pi$: $V = \sum_{i=1}^{N} V_i(s)$.

The *individual state-action value* for agent $i$ is defined as the expected discounted return of its own rewards:

$$Q_i(s, a_i) = \mathbb{E}\left[\sum_{t=0}^{\infty} \gamma^t R_i(s^t, a_i^t, a_{-i}^t) \mid s^0 = s, a_i^0 = a_i, a_{-i}^0 = a_{-i}, \boldsymbol{a}^t \sim \pi\right]. \tag{3}$$

The *joint state-action value* is the expected discounted return of the total rewards across all agents:

$$Q_{\text{tot}}(s, a) = \mathbb{E}\left[\sum_{t=0}^{\infty} \gamma^t \left(\sum_{i=1}^{N} R_i(s^t, \boldsymbol{a}^t)\right) \mid s^0 = s, \boldsymbol{a}^0 = \boldsymbol{a}, \boldsymbol{a}^t \sim \pi\right]. \tag{4}$$

While in the mechanism design setting, each agent $i$ maximizes expected cumulative mechanism-based reward:

$$\bar{V}_i(s) = \mathbb{E}\left[\sum_{t=0}^{\infty} \gamma^t \bar{R}_i(s^t, \boldsymbol{a}^t, R) \mid s^0 = s, \boldsymbol{a}^t \sim \pi\right]. \tag{5}$$

with the corresponding state–action value defined as:

$$\bar{Q}_i(s, a_i) = \mathbb{E}\left[\sum_{t=0}^{\infty} \gamma^t \bar{R}_i(s^t, a_i^t, a_{-i}^t, R) \mid s^0 = s, a_i^0 = a_i, a_{-i}^0 = a_{-i}, \boldsymbol{a}^t \sim \pi\right]. \tag{6}$$

By constructing rewards through $\bar{R}$, agents optimize their policies in a decentralized manner while aligning with the mechanism design objective. This setting induces the MDIE, where maximizing each agent's value function $\bar{V}_i$ is equivalent to maximizing social welfare.

**Assumption 3.1.** We assume that the joint state-action value $Q_{\text{tot}}$ can be factorized into $\bar{Q}_i$ for $i = 1, ..., N$ as

$$Q_{\text{tot}}(s, \boldsymbol{a}) = F(\bar{Q}_1(s, a_1), \bar{Q}_2(s, a_2), \ldots, \bar{Q}_N(s, a_N)), \tag{7}$$

where $F(\cdot)$ is a mixing function that preserves the consistency between $Q_{\text{tot}}$ and reshaped $\bar{Q}_i$.

**Definition 3.2** (Mechanism Design Induced Equilibrium). MDIE is an equilibrium in which the joint strategy $\pi^* = \{\pi_1^*, \pi_2^*, \ldots, \pi_N^*\}$ represents the set of optimal policies that maximizes each agent's mechanism design value function. Formally, the optimal policy for agent $i$ is defined as:

$$\pi_i^*(\cdot \mid s) = \arg\max_{\pi_i} \mathbb{E}\left[\sum_{t=0}^{\infty} \gamma^t \bar{R}_i(s^t, a_i^t, a_{-i}^t, R) \mid s^0 = s, \boldsymbol{a}^t \sim \pi\right], \tag{8}$$

where $\bar{R}_i(s, a_i, a_{-i}, R)$ denotes the mechanism design reward for agent $i$. That is, no agent can improve its expected return by unilaterally deviating from $\pi^*$ under the induced rewards.

**Lemma 3.3** (Existence of MDIE). *Under the MDMG framework, there exists at least one MDIE, which is a joint policy $\pi^*$ such that every agent's policy maximizes its expected mechanism design reward.*

The detailed proof is provided in Appendix A.1.

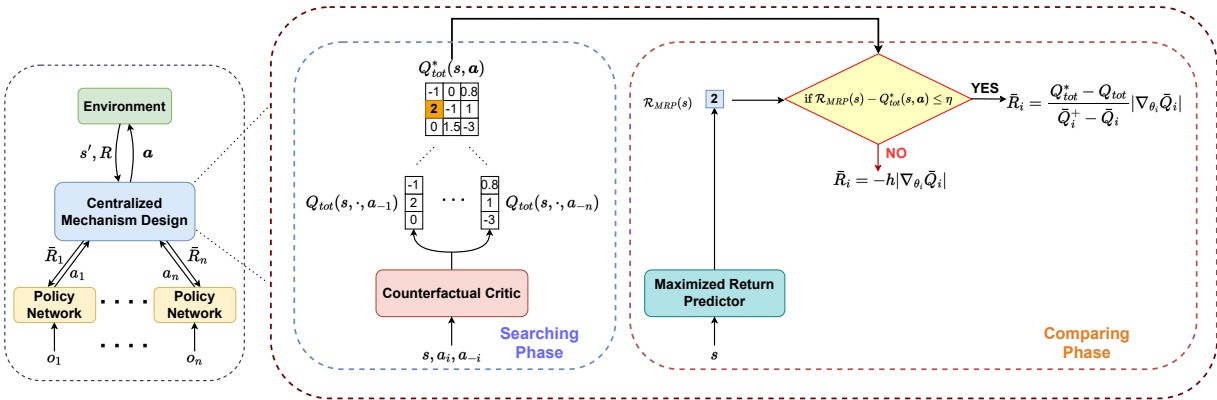

Figure 2: Overall framework of the proposed mechanism design approach. At each timestep $t$, taking agent $i$ as an example, the agent observes $o_i$ and outputs an action $a_i$ through its policy network. After collecting all agents' actions, the centralized mechanism design module reshapes the rewards $\bar{R}_i$ for each agent based on the global transition $(s, \mathbf{a}, s', R)$. The process includes two main phases: **(1) Searching Phase:** the Counterfactual Critic iterates $Q_{\text{tot}}(s, \cdot, a_{-i})$ by fixing other agents' actions $a_{-i}$, and finds the maximum value $Q_{\text{tot}}^*(s, \mathbf{a})$. **(2) Comparing Phase:** the MRP estimates a reference value $R_{\text{MRP}}(s)$. If $Q_{\text{tot}}^*(s, \mathbf{a}) - R_{\text{MRP}}(s) > \eta$, the mechanism assigns $\bar{R}_i = \frac{Q_{\text{tot}}^* - Q_{\text{tot}}}{\bar{Q}_i^+ - \bar{Q}_i} \cdot |\nabla_{\theta_i} \bar{Q}_i|$; otherwise, a penalty reward $\bar{R}_i = -h|\nabla_{\theta_i} \bar{Q}_i|$ is applied to encourage exploration.

## 3.2 Mechanism Design

To achieve the MDIE, we introduce a centralized mechanism that realigns these self-interested objectives by reshaping individual rewards so that each agent's optimal strategy simultaneously promotes social welfare. The key insight behind this mechanism is to explicitly measure how each agent's own parameters affect social welfare, and to use this dependency to construct a reward signal that bridges individual state-action value updates and welfare optimization.

To quantify how each agent's individual parameters affect the overall social welfare, we first examine the sensitivity of the joint state-action value $Q_{\text{tot}}$ with respect to the parameters $\theta_i$ of agent $i$'s state-action value function $\bar{Q}_i$. Intuitively, $\theta_i$ controls how agent $i$ selects actions or estimates its own value, and thus changing $\theta_i$ may indirectly affect social welfare. Using the chain rule, we design a heuristic mechanism that translates this dependency into a reshaped reward signal:

$$\bar{R}_i = \nabla_{\theta_i} Q_{\text{tot}} = \nabla_{\bar{Q}_i} Q_{\text{tot}} \cdot \nabla_{\theta_i} \bar{Q}_i, \approx \frac{Q_{\text{tot}}^* - Q_{\text{tot}}}{\bar{Q}_i^+ - \bar{Q}_i} \cdot \left| \nabla_{\theta_i} \bar{Q}_i \right|. \tag{9}$$

Here, $Q_{\text{tot}}^*$ represents the optimal joint state–action value, while $\bar{Q}_i^{(+)}$ denotes the corresponding individual value of agent $i$ evaluated under the same action configuration that achieves $Q_{\text{tot}}^*$. The action $a^*$ is obtained by enumerating the action space to find the one that yields the highest global value:

$$a^* = \arg\max_{a \in \mathcal{A}} Q_{\text{tot}}(s, a_i^*, a_{-i}^*) \tag{10}$$

Therefore, the $Q_{\text{tot}}^*$ and $\bar{Q}_i^+$ are defined as:

$$Q_{\text{tot}}^* = Q_{\text{tot}}(s, a_i^*, a_{-i}^*), \quad \bar{Q}_i^+ = \bar{Q}_i(s, a_i^*). \tag{11}$$

Eq. 9 captures the dynamic interaction between individual actions and their impact on social welfare. The ratio $\frac{Q_{\text{tot}}^* - Q_{\text{tot}}}{\bar{Q}_i^+ - \bar{Q}_i}$ quantifies the extent to which the policy change of agent $i$ affects the collective performance. As $Q_{\text{tot}}^* - Q_{\text{tot}}$ is non-negative, $\bar{Q}_i^+ - \bar{Q}_i$ determines whether the agent's policy change aligns with social

welfare. When $\bar{Q}_i^+ - \bar{Q}_i > 0$, individual and joint state-action values move in the same direction, indicating alignment. In contrast, when $\bar{Q}_i^+ - \bar{Q}_i < 0$, they diverge, indicating misalignment. The gradient term $|\nabla_{\theta_i} \bar{Q}_i|$ scales the adjustment according to the sensitivity of agent $i$'s value to the incentive. This adjustment ensures the reward signal reflects the evolving dynamics of the agent's policy. Because the adjustment depends only on the gradient of the current agent and not on access to other agents' internal parameters, the mechanism remains tractable as the number of agents grows.

**Counterfactual Critic Searching**   In the multi-agent setting, the joint action space grows exponentially with the number of agents $(O(|\mathcal{A}|^N))$, making it intractable to derive the mechanism reward $\bar{R}$ in Eq. 9 by computing the optimal joint action $a^*$. To overcome this challenge, the first stage of our mechanism design, illustrated in Fig. 2, performs a *counterfactual search* that reduces the complexity to $O(N|\mathcal{A}|)$. Specifically, agent $i$ is activated while keeping the others' actions fixed, and the centralized mechanism evaluates how the deviation of agent $i$'s action affects the joint state-action value function $Q_{\text{tot}}(s, \boldsymbol{a})$.

The counterfactual critic computes $Q_{\text{tot}}(s, \boldsymbol{a})$ and approximates the optimal $Q_{\text{tot}}^*(s, \boldsymbol{a})$ through an iterative best-response search, where each agent is activated in turn while the others' actions remain fixed. For each agent $i$, the critic evaluates $Q_{\text{tot}}(s, \boldsymbol{a})$ across all possible actions, $Q_{\text{tot}}(s, a_i, a_{-i})$ for $a_i \in \mathcal{A}_i$, and updates the joint action by selecting $a_i^*$ that maximizes $Q_{\text{tot}}(s, \boldsymbol{a})$:

$$Q_{\text{tot}}^*(s, \boldsymbol{a}) \approx \max \left[ Q_{\text{tot}}\big(s, \cdot, a_{-1}\big), \ \ldots, \ Q_{\text{tot}}\big(s, \cdot, a_{-N}\big) \right], \tag{12}$$

where each term $Q_{\text{tot}}(s, \cdot, a_{-i})$ denotes the slice of joint state-action values evaluated by agent $i$'s action, with the actions for other agents fixed.

Importantly, this centralized mechanism containing the counterfactual critic is only used during *training* to produce the reshaped reward signals $\bar{R}_i$ for each agent to update their individual value $\bar{Q}_i$. The training procedure is similar to (Foerster et al., 2018), the $\psi$-parameterized counterfactual critic is optimized by minimizing the temporal-difference (TD) loss:

$$\mathcal{L}_{\text{critic}} = \mathbb{E}\left[ \left( Q_{\text{tot}}^{\psi}(s, \boldsymbol{a}) - (R + \gamma \max_{\boldsymbol{a}'} Q_{\text{tot}}^{\psi}(s', \boldsymbol{a}'))\right)^2 \right]. \tag{13}$$

where the $s'$ and $\boldsymbol{a}'$ indicate the state and the action of the next step, respectively.

**Comparing Phase with Maximized Return Predictor**   Although the counterfactual critic network can evaluate hypothetical deviations in an agent's action to improve the joint policies, it may become trapped in a local optimum. This limitation arises because global optimality can only be achieved if the fixed actions $a_{-i}$ of the other agents are already optimal. In practice, however, these actions are generated by agents that learn simultaneously, leading to potential convergence to suboptimal equilibria.

To mitigate the local-optimum limitation of the counterfactual critic, we introduce a *comparing phase* equipped with a *MRP*. The MRP estimates the maximum achievable returns under optimal conditions (Blundell et al., 2016; Pritzel et al., 2017; Zheng et al., 2021). Instead of relying on direct search results, the MRP predicts the maximum possible joint value $\mathcal{R}_{\text{MRP}}(s)$ for each state $s$, serving as a reference for evaluating whether the agents' current joint actions approach the optimal region. A dynamic buffer $\mathcal{R}_{\text{MRP}}$ stores state and its associated maximum returns, updated according to the following rule:

$$\mathcal{R}_{\text{MRP}}(s) = \begin{cases} \max\{\mathcal{R}_{\text{MRP}}(s), Q_{tot}(s, \boldsymbol{a})\}, & \text{if } s \in \text{dom}(\mathcal{R}_{\text{MRP}}) \\ Q_{tot}(s, \boldsymbol{a}), & \text{otherwise} \end{cases} \tag{14}$$

By iteratively updating the dynamic buffer, the MRP estimates the maximum possible value of $Q_{tot}$ for a given state. The MRP is designed not only to retrieve the highest value encountered for known states but also to generalize this estimation to predict maximum values for previously unseen states.

We then integrate the MRP into the mechanism design framework in the comparing phase (see Fig. 2). The mechanism evaluates whether the agents' joint state-action value $Q_{\text{tot}}(s, \boldsymbol{a})$ is close to the predicted maximum $\mathcal{R}_{\text{MRP}}(s)$. If the difference exceeds a predefined threshold $\eta$, the agents are considered to be

near a suboptimal solution, and a penalty factor $h$ is applied to encourage policy exploration. The revised mechanism design reward is given by:

$$\bar{R}_i = \begin{cases} \frac{Q_{\text{tot}}^* - Q_{\text{tot}}}{\bar{Q}_i^+ - \bar{Q}_i} \cdot \left| \nabla_{\theta_i} \bar{Q}_i \right|, & \text{if } \mathcal{R}_{\text{MRP}}(s) - Q_{\text{tot}}^*(s, \boldsymbol{a}) \leq \eta, \\ -h \cdot \left| \nabla_{\theta_i} \bar{Q}_i \right|, & \text{if } \mathcal{R}_{\text{MRP}}(s) - Q_{\text{tot}}^*(s, \boldsymbol{a}) > \eta, \end{cases} \tag{15}$$

The parameters $\eta$ and $h$ are hyperparameters of the MRP module. Their values are determined empirically through grid search and validated across multiple scenarios.

The MRP network is trained to regress these stored maximum values using a mean-squared error (MSE) loss:

$$\mathcal{L}_{\text{MRP}} = \mathbb{E}_{s \sim \mathcal{D}} \left[ \left( \mathcal{R}_{\text{MRP}}(s) - \hat{\mathcal{R}}_{\text{MRP}}^{\phi}(s) \right)^2 \right], \tag{16}$$

where $\hat{\mathcal{R}}_{\text{MRP}}^{\phi}(s)$ denotes the predicted return parameterized by $\phi$, and $\mathcal{D}$ is the replay buffer.

**Lemma 3.4** (Monotonicity and Contraction of the MRP Operator). *Let $Q_{\text{tot}}$ denote the joint state-action value estimated, and let the MRP operator be defined as*

$$(\mathcal{M}Q)(s) = \max_{\boldsymbol{a}} Q_{\text{tot}}(s, \boldsymbol{a}), \tag{17}$$

*for each state $s \in \mathcal{S}$. Then the operator $\mathcal{M}$ satisfies:*

1. ***Monotonicity:*** *For any two functions $Q$ and $Q'$ with $Q(s, \boldsymbol{a}) \leq Q'(s, \boldsymbol{a})$ for all $(s, \boldsymbol{a})$, it follows that $(\mathcal{M}Q)(s) \leq (\mathcal{M}Q')(s)$ for all $s$.*

2. ***Contraction:*** *Under the standard discounted setting with factor $\gamma \in (0, 1)$, the associated Bellman operator with MRP is a $\gamma$-contraction in the sup-norm, i.e.,*

$$\|\mathcal{T}_{\mathcal{M}}Q - \mathcal{T}_{\mathcal{M}}Q'\|_{\infty} \leq \gamma \|Q - Q'\|_{\infty},$$

*where $\mathcal{T}_{\mathcal{M}}$ denotes the Bellman operator using $\mathcal{R}_{MRP}(s) = (\mathcal{M}Q)(s)$ as the value estimate.*

The detailed proof is provided in Appendix A.2.

### 3.3 Value Iteration Algorithm

After establishing the reshaped reward signal $\bar{R}_i$, the next step is to ensure convergence to the MDIE introduced in definition 3.2. To this end, we develop a tailored *value iteration algorithm* that iteratively updates each agent's individual value network under the reshaped rewards. This algorithm aligns individual state-action value updates with mechanism design reward and provides convergence guarantees. The following section presents the update rules and theoretical analysis.

First of all, it is essential to define the Bellman-type equation:

$$T_i(s, \boldsymbol{a}) = \bar{R}_i(s, \boldsymbol{a}) + \gamma \sum_{s' \in S} P(s' \mid s, \boldsymbol{a}) \bar{V}_i^*(s'), \tag{18}$$

Then the Bellman-type equation can be written as:

$$\bar{V}_i^*(s) = \max_{\pi_i(\cdot | s)} \sum_{\boldsymbol{a} \in A} \pi_i(a_i \mid s) \prod_{j \neq i} \pi_j(a_j \mid s) T_i(s, \boldsymbol{a}), \tag{19}$$

$$= \max_{\pi_i(\cdot | s)} \mathbb{E}_{a_i \sim \pi_i(\cdot | s), a_{-i} \sim \pi_{-i}^*(\cdot | s)} \left[ T_i(s, \boldsymbol{a}) \right]. \tag{20}$$

Eq. 20 forms the basis for value iteration. The update rule is defined as

$$\bar{V}_i^{t+1}(s) = \max_{\pi_i(\cdot | s)} \sum_{\boldsymbol{a} \in A} \pi_i(a_i \mid s) \prod_{j \neq i} \pi_j(a_j \mid s) T_i(s, \boldsymbol{a}). \tag{21}$$

Building on Eq. 21, a Q-learning-based formulation can be derived. The equilibrium action–value function satisfies the Bellman equation

$$\bar{Q}_i^* \left(s, \boldsymbol{a}, \bar{R}_i(s, \boldsymbol{a}, R)\right) = \bar{R}_i(s, \boldsymbol{a}, R) + \gamma \sum_{s' \in S} P(s'|s, \boldsymbol{a}) \times \sum_{\boldsymbol{a}'} \left( \prod_{j=1}^{N} \pi_j^*(a_j'|s') \right) \bar{Q}_i^* \left(s', \boldsymbol{a}', \bar{R}^i(s', \boldsymbol{a}', R')\right). \quad (22)$$

Following Mnih et al. (2013), we parameterize the equilibrium action–value function $\bar{Q}_i(s, \boldsymbol{a}, \bar{R}_i)$ using a neural network $\bar{Q}_i(s, \boldsymbol{a}, R; \theta_i)$ with learnable parameters $\theta_i$. The corresponding TD target for agent $i$ at transition $(s, \boldsymbol{a}, \bar{R}_i, s')$ is defined as

$$y_i(s, \boldsymbol{a}, s') = \bar{R}_i(s, \boldsymbol{a}, R) + \gamma \, \mathbb{E}_{\boldsymbol{a}' \sim \prod_j \pi_j^*(\cdot|s')} \left[ \bar{Q}_i(s', \boldsymbol{a}', R'; \theta_i^-) \right], \quad (23)$$

where $\theta_i^-$ denotes the parameters of a target network used for stable training. The network parameters $\theta_i$ are optimized by minimizing the MSE loss over a minibatch $\mathcal{B}$:

$$\mathcal{L}_{Q_i}(\theta_i) = \frac{1}{|\mathcal{B}|} \sum_{(s, \boldsymbol{a}, \bar{R}_i, s') \in \mathcal{B}} \ell\big( y_i(s, \boldsymbol{a}, s') - \bar{Q}_i(s, \boldsymbol{a}; \theta_i)\big), \quad (24)$$

where $\ell(\cdot)$ denotes the MSE loss function. The detailed training procedure is summarized in Algorithm 1.

**Theorem 3.5.** *Under the conditions of the compactness of the state and action spaces, the boundedness and continuity of the reward and transition functions, the optimal state-action value function $\bar{Q}_i^*$, as defined by the Bellman equation, is guaranteed to exist and is unique.*

The detailed proof can be found in Appendix A.3.

**Theorem 3.6** (Optimality of MDIE)**.** *Under the MDMG framework with a counterfactual critic and a maximized return predictor, the MDIE coincides with the globally optimal joint policy that maximizes the joint state-action value $Q_{\text{tot}}$.*

The detailed proof is listed in Appendix A.4.

## 4 Experimental Results

In this section, we present empirical evaluations to demonstrate the validity and effectiveness of the proposed method. Specifically, we aim to address the following questions:

1. **Performance Evaluation:** How does our mechanism design algorithm perform across different settings?

2. **Reward Function Reshaping:** Can the proposed algorithm effectively and accurately compute the mechanism design reward function for each agent?

3. **MRP Examination:** How important is the MRP module, and what is the impact of removing it from the framework?

4. **Fairness Analysis:** How does the proposed mechanism affect the fairness of individual agents while improving social welfare? (Discussed in Appendix B.1)

### 4.1 Environment Settings and Baselines

To validate the effectiveness of the proposed framework, we conduct experiments in four representative environments that capture different aspects of social dilemmas and welfare challenges: **Escape Room** (Yang et al., 2020), **Multi-Agent Bidding**, **Bandwidth Allocation in Communication Channels** (Parvini et al., 2023), and **Harvest** (Vinitsky et al., 2019). These environments are commonly used in the literature to illustrate the conflict between individual self-interest and social welfare.

### 4.1.1 Environment Settings

**Escape Room** Escape Room is a commonly used $N$-player Markov game that requires explicit social welfare for success (Yang et al., 2020). The game consists of three distinct states: the *initial state*, the *lever state*, and the *door state*. All agents start from the initial state and aim to reach the door, which represents the terminal state. The door opens only if at least $M$ agents simultaneously pull the lever, enabling the rest of the agents to exit. Pulling the lever incurs an individual cost of $-1$, while agents who successfully exit through the door obtain a reward of $+10$. We classify agents that pull the lever as *cooperators*, while agents who escape are considered *winners*. This environment introduces a fundamental social dilemma: although social welfare is necessary to maximize joint state-action value, each agent has an incentive to free-ride on others' effort and avoid the lever cost. To further examine scalability, we design multiple configurations with different group sizes, specifically $(N = 5, M = 3)$, $(N = 8, M = 5)$, and $(N = 10, M = 6)$.

**Multi-Agent Bidding** The Multi-Agent Bidding environment models a competitive auction among multiple agents bidding for a single item of hidden value $c_h$. Each agent $i$ submits a bid $b_i$ and incurs a bidding cost $c_i$. The highest bidder wins the item, but the associated cost is subtracted from their environmental rewards, resulting in a tension between aggressive competition and collective efficiency. While each agent is motivated to outbid others to maximize its own reward, uncoordinated competition leads to resource waste and negative collective returns. The socially optimal outcome occurs when only one agent bids and others abstain, minimizing redundant expenditures—a clear example of a social dilemma where individual rationality harms overall welfare.

**Bandwidth Allocation in Communication Channels** We further evaluate our framework in a real-world–inspired *Bandwidth Allocation* task, adapted from Parvini et al. (2023) and Wang et al. (2025). In this environment, $N$ connected autonomous vehicles share $C$ communication channels to transmit information while maintaining string stability. Each agent selects its transmission power and channel assignment to maximize its own data rate:

$$R_i = k_1 B \log_2 \left( 1 + \frac{P_i h_i}{\sigma^2 + \sum_{j \neq i} P_j h_j} \right), \tag{25}$$

where $B$ is the channel bandwidth, $P_i$ is the transmit power, $h_i$ is the channel gain, and $\sigma^2$ is the noise power. The denominator term represents interference from other agents, indicating that as more vehicles transmit on the same channel, the signal-to-interference-plus-noise ratio (SINR) deteriorates. As each agent increases its transmission power to improve its own data rate, it simultaneously raises interference for others, causing everyone's communication quality to deteriorate—an inherent social dilemma in shared wireless systems.

**Harvest** Harvest is a canonical sequential social dilemma environment that captures the tension between short-term individual gain and long-term collective welfare (Vinitsky et al., 2019). In this grid-based environment, multiple agents navigate a shared map to collect renewable resources (apples). Each agent receives a positive reward for harvesting an apple, while the regeneration rate of apples depends on the local density of remaining apples. Specifically, excessive harvesting in a local region reduces the probability of future apple spawning, potentially leading to resource depletion. As a result, agents face a dilemma between aggressively collecting apples to maximize immediate individual rewards and restraining their behavior to preserve long-term collective returns. Each agent observes a local grid centered at its current position, and the action space is discrete, including movement in four directions or staying in place. To facilitate centralized decision-making, we construct a centralized state by concatenating the individual observations of all agents.

### 4.1.2 Baselines

In our experiments, we compare our proposed mechanism design-based algorithm (MD) against four representative baselines: Independent Q-Learning (IQL), Centralized Training (CEN) (Rashid et al., 2018), Incentive Design (ID) (Yang et al., 2021), and Learning to Incentivize Others (LIO) (Yang et al., 2020). IQL serves as the fully decentralized baseline where each agent learns its own $Q$-function solely based on its individual reward signal from the environment, without access to global information or coordination. CEN represents a centralized training paradigm, same as (Rashid et al., 2018), where a single global reward

signal is used to train a joint $Q$-network for all agents. ID belongs to the line of incentive design methods, where agents shape rewards by modeling the impact of their actions on others. This requires access to other agents' internal parameters to help model the reward signals for the training. LIO follows the idea of opponent modeling: each agent learns to build models of other agents' policies and uses them to generate incentive rewards.

## 4.2 Simulation Results

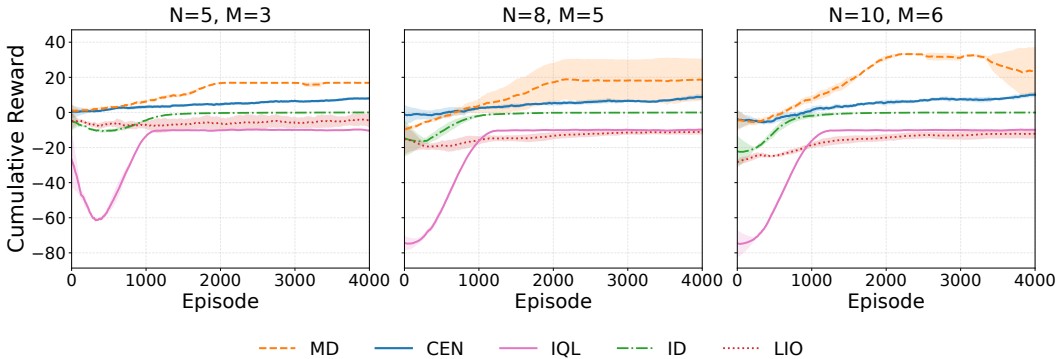

Figure 3: Escape room experiment results.

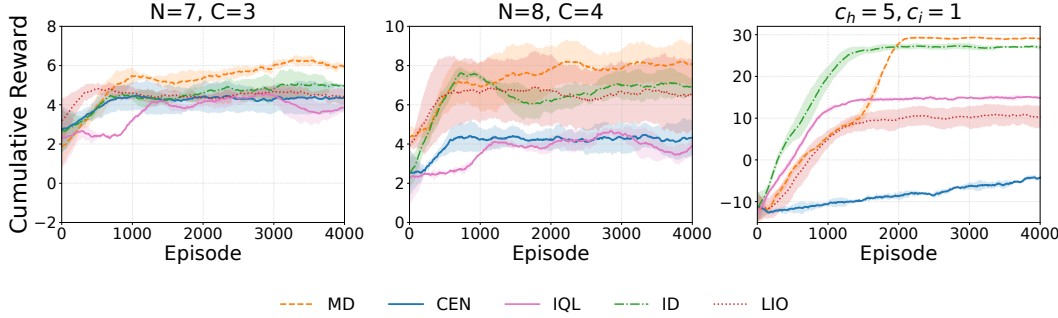

Figure 4: Resource allocation in communication scenarios and multi-agent bidding results. Learning curves of cumulative rewards under increasing strategic complexity. **Left and middle**: communication resource allocation with different numbers of agents and channel contention levels ($N = 7, M = 3$ and $N = 8, M = 4$). **Right**: multi-agent bidding with hidden item value ($c_h = 5, c_i = 1$), where uncoordinated competitive bidding leads to inefficient outcomes.

### 4.2.1 Performance Evaluation

We evaluate the performance of our proposed mechanism design-based algorithm against several representative baselines across multiple environments and settings. The learning curves in Fig. 3, 4 and 5 report the mean cumulative reward at different seeds, with shaded areas denoting the variance.

These figures show that our method achieves superior performance. In the simplest case ($N = 5, M = 3$) of escape room settings, MD quickly learns to allocate several agents to the lever while other agents to the door, converging to the global optimum. In contrast, IQL fails due to the absence of coordination: each agent greedily maximizes its own return and thus often leaves the lever or initial state unattended. CEN also struggles since training with a single global reward signal leads to poor credit assignment, especially for non-monotonic settings, making it difficult for individual agents to discern their contributions. ID and LIO provide partial improvements by introducing incentive mechanisms or opponent modeling, but they both rely on access to other agents' internal parameters and focus on shaping rewards without directly aligning with

the mechanism design reward. Consequently, both methods stagnate in sub-optimal social welfare modes. As the task complexity increases ($N = 8, M = 5$ and $N = 10, M = 6$), the performance gap further widens. MD maintains robust convergence to the optimal solution, while the baselines deteriorate: IQL collapses almost entirely, CEN saturates at low rewards, and both ID and LIO fail to scale effectively. These results highlight the scalability advantage of MD—its mechanism-driven reward transformation continues to guide agents toward the effective social welfare outcome even as the population size and coordination demands grow.

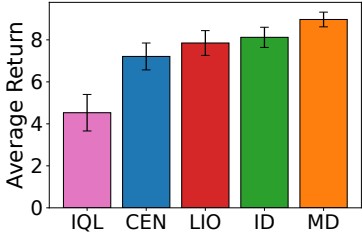

Figure 5: Harvest experiment results.

Similar patterns appear in the multi-agent bidding, bandwidth allocation, harvest tasks in Fig. 4 and Fig. 5. MD consistently achieves higher cumulative rewards and remains effective as the number of users grows. These results highlight that MD not only aligns local incentives with the mechanism design reward but also maintains robust scalability as the number of agents increases.h

### 4.2.2 Reward Function Reshaping

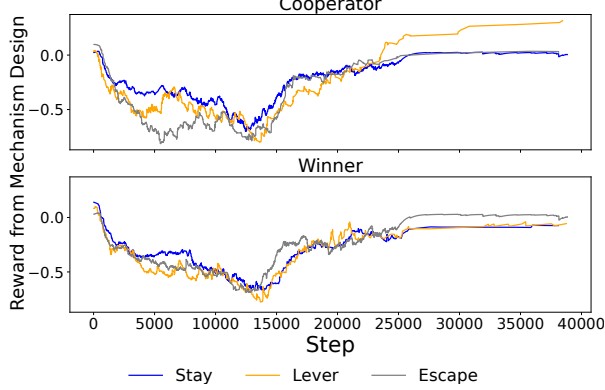
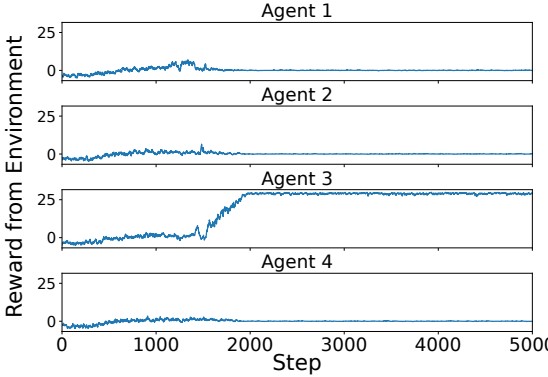

Figure 6: Mechanism Rewards in Escape Room ($N = 5, M = 3$).

Figure 7: Each Agent's Environment Reward in Multi-Agent Bidding.

Fig. 6 shows the mechanism rewards in the Escape Room setting with $N = 5$ and $M = 3$. The first row and second row correspond to the cooperator role and the winner role, respectively. The different curves represent the reward trajectories for the actions *Stay*, *Lever*, and *Escape*. In the early stage of training, the rewards for different actions overlap, and agents tend to behave selfishly. As training progresses, the mechanism gradually amplifies the gap: cooperative actions, such as pulling the lever or moving toward the exit, receive increasingly higher rewards, while non-cooperative actions become less valuable. Importantly, the incentives for cooperators are reshaped so that cooperation behaviors dominate over inaction, and the winner's reward for escaping becomes positive only when the lever has been pulled, thereby binding self-interested behavior to collective social welfare. This evolution shows how the mechanism reshapes rewards over time, aligning individual goals with social welfare.

Fig. 7 illustrates each agent's environment reward trends in the multi-agent bidding task. As training progresses, a clear division of roles emerges: three agents gradually reduce their bidding activity and converge

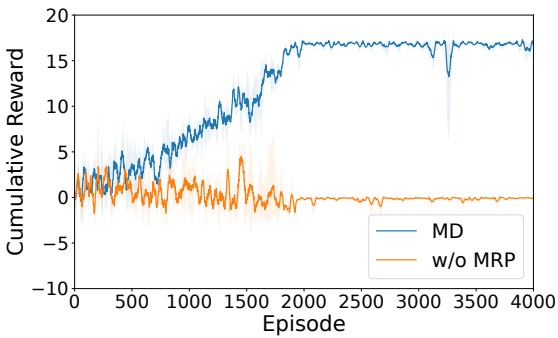

Figure 8: Ablation Results in Escape Room N=5, M=3.

to near-zero rewards, while a single agent (Agent 3) consistently secures the item at the lowest price and maintains a stable positive payoff. This outcome indicates that the mechanism design successfully discourages inefficient competition, as the non-bidding agents learn that continued bidding only reduces their rewards without yielding additional gains. From a broader perspective, the mechanism-derived rewards restructure the game into a stable equilibrium where redundancy is eliminated and resources are allocated efficiently.

### 4.2.3  MRP Examination

To evaluate the importance of MRP module, we conduct an ablation study in the Escape Room environment with $N = 5$ and $M = 3$. Fig. 8 compares the full MD framework against its variant without MRP (denoted as w/o MRP). The results show a clear contrast: while MD steadily improves and converges to the global optimum, removing MRP causes learning to stagnate at a much lower reward level, with high variance across episodes.

This difference can be attributed to how MRP reshapes the exploration space. Without MRP, the mechanism only updates reward based on single-agent deviations while holding others fixed, which easily traps the system in suboptimal equilibria. In contrast, MRP propagates the mechanism design rewards across the joint action space, ensuring that agents receive informative feedback about the impact of their coordinated behaviors. This makes it possible to escape poor equilibria and converge to the globally optimal strategy. For instance, consider the case where the agents choose the joint action *(lever, stay, stay, stay, stay)*. In principle, the globally optimal solution should be *(lever, lever, escape, escape, escape)*, where two agents pull the lever and three agents escape. However, without MRP, the counterfactual critic only evaluates single-agent deviations while keeping the others fixed. Under this restricted update, the system may incorrectly conclude that switching the first agent from *lever* to *stay* improves the collective return, since *(stay, stay, stay, stay, stay)* can temporarily yield a higher immediate return than the partially cooperative configuration. As a result, the learning dynamics converge toward all agents choosing *stay*, which is a clear suboptimal equilibrium. In contrast, MRP propagates mechanism-adjusted feedback across joint actions, ensuring that cooperative lever-pulling behaviors are reinforced rather than eliminated, thus guiding the system toward the true global optimum.

## 5  Conclusion

This paper introduced the MDMG, a framework designed to resolve social dilemmas in self-interested multi-agent systems. By leveraging mechanism design to dynamically reshape agents' utility functions, our method aligns self-interested learning with social welfare without requiring access to other agents' internal parameters or centralized control. The proposed framework establishes the MDIE and provides theoretical guarantees of its existence and optimality. Our value iteration algorithm, integrated with a counterfactual critic and a maximized return predictor, enables agents to converge toward equilibria that improve social welfare. Extensive experiments in escape room, multi-agent bidding, and bandwidth allocation tasks demonstrate that our method effectively mitigates selfish behavior and achieves higher social welfare.

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

# A Mathematical Proof

## A.1 Proof of Lemma 3.3

*Proof.* We aim to prove the existence of the MDIE by leveraging Kakutani's fixed-point theorem. The assumptions required are as follows:

**Assumptions:**

- **Compact state and action spaces:** The state space $\mathcal{S}$ and action space $\mathcal{A}$ are compact.

- **Non-empty, convex, and compact strategy space:** The strategy set $\Pi$ is non-empty, convex, and compact. The individual strategy space for each agent $i$ is defined as:

$$\Pi_i = \{\pi_i \mid \pi_i(a_i|s) \geq 0, \sum_{a_i} \pi_i(a_i|s) = 1, \forall s \in S\}.$$

  To show convexity, let $\pi_i, \pi_i' \in \Pi_i$ be two valid policies. Define a convex combination:

$$\tilde{\pi}_i = \lambda \pi_i + (1 - \lambda)\pi_i', \quad \text{for } \lambda \in [0, 1].$$

  Then, for any $s \in S$ and $a_i \in A_i$,

$$\sum_{a_i} \tilde{\pi}_i(a_i|s) = \sum_{a_i}(\lambda \pi_i(a_i|s) + (1 - \lambda)\pi_i'(a_i|s))$$
$$= \lambda \sum_{a_i} \pi_i(a_i|s) + (1 - \lambda) \sum_{a_i} \pi_i'(a_i|s) = 1.$$

  Since $\tilde{\pi}_i(a_i|s) \geq 0$ for all $a_i$, we conclude that $\tilde{\pi}_i \in \Pi_i$, proving that $\Pi_i$ is convex.

  Since the global strategy space is the Cartesian product of individual strategy spaces,

$$\Pi = \Pi_1 \times \Pi_2 \times \cdots \times \Pi_N,$$

  and the Cartesian product of convex sets remains convex, $\Pi$ is also convex.

- **Value function uniqueness:** The optimal value function $V^*$ and the optimal action-value function $Q^*$ are unique, as established in previous results.

- **Continuity:** The mappings $V^*$ and $Q^*$ are continuous with respect to the state $s$ and action $\boldsymbol{a}$.

Define the set-valued mapping $F : \mathcal{S} \to 2^{\Pi}$ as:

$$F(s) = \{\pi \in \Pi \mid \pi(s) \in \mathcal{A}^*(s)\},$$

where $\mathcal{A}^*(s)$ is the set of optimal actions at state $s$.

Since $F$ satisfies all the conditions of Kakutani's fixed-point theorem, there exists a fixed point $\pi^* \in \Pi$ such that:

$$\pi^*(s) \in \mathcal{A}^*(s), \quad \forall s \in \mathcal{S}.$$

This fixed point $\pi^*$ represents the MDIE. Hence, the existence of MDIE is guaranteed under the given assumptions.

$\square$

## A.2 Proof of Lemma 3.4

*Proof.* **Monotonicity.** Suppose two value functions $Q$ and $Q'$ satisfy $Q(s,a) \le Q'(s,a)$ for all $(s,a) \in \mathcal{S} \times \mathcal{A}$. For any fixed state $s$, this implies that each action-value under $Q$ is no greater than the corresponding one under $Q'$. Taking the maximum over the action space preserves this ordering, i.e.,

$$(MQ)(s) = \max_a Q(s,a) \ \le \ \max_a Q'(s,a) = (MQ')(s).$$

Thus, the operator $M$ is monotone. Also, the update rule of the maximized return predictor ensures monotonicity. Since $\mathcal{RMRP}(s)$ is updated as the maximum between its previous value and the newly observed joint return $Qtot(s,\boldsymbol{a})$, it follows that $\mathcal{R}_{\mathrm{MRP}}(s)$ is non-decreasing over time. This property directly implies the monotonicity of the operator, consistent with the max-Bellman operator analyzed in Blundell et al. (2016).

**Contraction.** Consider the Bellman operator with the maximized return predictor:

$$(\mathcal{T}_{\mathcal{M}}Q)(s,a) = R(s,a) + \gamma \, \mathbb{E}_{s'}\big[(\mathcal{M}Q)(s')\big].$$

Let $Q, Q'$ be two value functions. Then

$$\big|(\mathcal{T}_{\mathcal{M}}Q)(s,a) - (\mathcal{T}_{\mathcal{M}}Q')(s,a)\big| = \gamma\Big|\mathbb{E}_{s'}\big[(\mathcal{M}Q)(s') - (\mathcal{M}Q')(s')\big]\Big|$$
$$\le \gamma \, \|Q - Q'\|_\infty.$$

Taking the supremum over all $(s,a)$ yields

$$\|\mathcal{T}_{\mathcal{M}}Q - \mathcal{T}_{\mathcal{M}}Q'\|_\infty \le \gamma \, \|Q - Q'\|_\infty,$$

showing $\mathcal{T}_{\mathcal{M}}$ is a $\gamma$-contraction.

By Banach's fixed point theorem, $\mathcal{T}_{\mathcal{M}}$ converges to a unique fixed point $Q^*_{\mathrm{MRP}}$, which corresponds to the optimal maximized return predictor. $\qquad\square$

## A.3 Proof of Theorem 3.5

*Proof.* We first list the assumptions necessary to prove the contraction mapping property of the Bellman operator:

**Assumptions:**

- **Bounded reward:** The reward function $\bar{R}_i(s,\boldsymbol{a})$ is bounded for all $s \in \mathcal{S}$ and $\boldsymbol{a} \in \mathcal{A}$, i.e., $|\bar{R}_i(s,\boldsymbol{a})| \le R_{\max}$.

- **Bounded value function:** The Q-value functions $\bar{Q}_i(s,\boldsymbol{a})$ and $\bar{Q}'_i(s,\boldsymbol{a})$ are bounded for all $s \in \mathcal{S}$ and $\boldsymbol{a} \in \mathcal{A}$, i.e., $|\bar{Q}_i(s,\boldsymbol{a})| \le Q_{\max}$.

- **Discount factor:** The discount factor satisfies $0 \le \gamma < 1$.

- **Transition probabilities:** The transition probability $P(s'|s,\boldsymbol{a})$ satisfies $\sum_{s'} P(s'|s,\boldsymbol{a}) = 1$ for all $s \in \mathcal{S}$ and $\boldsymbol{a} \in \mathcal{A}$.

- **Compact state and action spaces:** The state space $\mathcal{S}$ and action space $\mathcal{A}$ are compact sets, ensuring that optimization problems over these spaces (e.g., $\max_{\boldsymbol{a}}$) have well-defined solutions.

- **Lipschitz property of** max: The max operator satisfies the 1-Lipschitz property:

$$\Big|\max_{\boldsymbol{a}'} \bar{Q}_i(s',\boldsymbol{a}') - \max_{\boldsymbol{a}'} \bar{Q}'_i(s',\boldsymbol{a}')\Big| \le \|\bar{Q}_i - \bar{Q}'_i\|_\infty.$$

To prove the algorithm can converge to a unique value, we have to show the Bellman operator $T$ is a contraction mapping under the following conditions.

Following the proof of Zhang et al. (2020) and Kardeş et al. (2011), we define the Bellman operator $T$ as:

$$T\bar{Q}_i(s, \boldsymbol{a}) = \bar{R}_i(s, \boldsymbol{a}) + \gamma \sum_{s'} P(s'|s, \boldsymbol{a}) \max_{\boldsymbol{a}'} \bar{Q}_i(s', \boldsymbol{a}').$$

To prove that $T$ is a contraction mapping, we analyze the difference $T\bar{Q}_i - T\bar{Q}_i'$ in two cases.

**Case 1:** $T\bar{Q}_i(s, \boldsymbol{a}) \geq T\bar{Q}_i'(s, \boldsymbol{a})$. We have:

$$
\begin{aligned}
T\bar{Q}_i(s, \boldsymbol{a}) - T\bar{Q}_i'(s, \boldsymbol{a}) &= \gamma \sum_{s'} P(s'|s, \boldsymbol{a}) \left( \max_{\boldsymbol{a}'} \bar{Q}_i(s', \boldsymbol{a}') - \max_{\boldsymbol{a}'} \bar{Q}_i'(s', \boldsymbol{a}') \right) \\
&\leq \gamma \sum_{s'} P(s'|s, \boldsymbol{a}) \sup_{s', \boldsymbol{a}'} \left| \bar{Q}_i(s', \boldsymbol{a}') - \bar{Q}_i'(s', \boldsymbol{a}') \right| \\
&= \gamma \sup_{s', \boldsymbol{a}'} \left| \bar{Q}_i(s', \boldsymbol{a}') - \bar{Q}_i'(s', \boldsymbol{a}') \right| \\
&= \gamma \| \bar{Q}_i - \bar{Q}_i' \|_\infty.
\end{aligned}
$$

**Case 2:** $T\bar{Q}_i(s, \boldsymbol{a}) \leq T\bar{Q}_i'(s, \boldsymbol{a})$. Similarly, we have:

$$T\bar{Q}_i'(s, \boldsymbol{a}) - T\bar{Q}_i(s, \boldsymbol{a}) \leq \gamma \| \bar{Q}_i - \bar{Q}_i' \|_\infty.$$

From both cases, we conclude that:

$$\| T\bar{Q}_i - T\bar{Q}_i' \|_\infty \leq \gamma \| \bar{Q}_i - \bar{Q}_i' \|_\infty,$$

where $\gamma \in [0, 1)$. Hence, $T$ is a contraction mapping.

By the Banach fixed-point theorem, since $T$ is a contraction mapping, there exists a unique fixed point $\bar{Q}_i^*$ such that:

$$T\bar{Q}_i^* = \bar{Q}_i^*.$$

This completes the proof. $\qquad\square$

### A.4   Proof of Theorem 3.6

*Proof.* We aim to show that the optimal joint policy derived from the mechanism design value function, denoted by $\pi^*$, coincides with the original globally optimal policy $\pi^*$ that maximizes $Q_{\text{tot}}$.

First, recall the mechanism-based update:

$$\bar{R}_i \;=\; \nabla_{\theta_i} Q_{\text{tot}} \;=\; \nabla_{\bar{Q}_i} Q_{\text{tot}} \;\cdot\; \nabla_{\theta_i} \bar{Q}_i \;\approx\; \frac{Q_{\text{tot}}^* - Q_{\text{tot}}}{\bar{Q}_i' - \bar{Q}_i} \;\cdot\; \left| \nabla_{\theta_i} \bar{Q}_i \right|.$$

From this relationship, we observe that $\nabla_{\theta_i} \bar{Q}_i$ is *proportional* to $\nabla_{\theta_i} Q_{\text{tot}}$ by a strictly positive factor. In other words, for every agent $i$,

$$\nabla_{\theta_i} Q_{\text{tot}} \;=\; g_i(\theta_1, \ldots, \theta_N) \, \nabla_{\theta_i} \bar{Q}_i, \quad \text{with } g_i(\theta_1, \ldots, \theta_N) > 0.$$

This proportional relationship implies that any policy update that increases $\bar{Q}_i$ also increases $Q_{\text{tot}}$ in the same direction in parameter space. Consequently, for a given state $s$ and the other agents' actions $\boldsymbol{a}_{-i}$, the local preference ordering over actions $a_i$ induced by $\bar{Q}_i(s, a_i, \boldsymbol{a}_{-i})$ is the same as that induced by $Q_{\text{tot}}(s, a_i, \boldsymbol{a}_{-i})$. Formally, if

$$Q_{\text{tot}}(s, a_i, \boldsymbol{a}_{-i}) \;>\; Q_{\text{tot}}(s, a_i', \boldsymbol{a}_{-i}),$$

then it follows that

$$\bar{Q}_i(s, a_i, \boldsymbol{a}_{-i}) \; > \; \bar{Q}_i(s, a_i', \boldsymbol{a}_{-i}).$$

Follow the previous MARL work (Rashid et al., 2018; Foerster et al., 2018), we assume a weighted decompositions to express the $Q_{tot}$ by $\bar{Q}_i$, followed by:

$$Q_{\text{tot}}(s, \boldsymbol{a}) \; = \; \sum_i w_i(s)\, Q_i(s, a_i) \; + \; b(s),$$

And in other words, if $\pi_{tot}^{new} = \{\pi_i^{new}, \pi_{-i}^{old}\}$ differs from $\pi_{tot}^{old}$ only by agent $i$'s new policy, we have:

$$\mathbb{E}_{\pi_{tot}^{new}}\left[Q_{\text{tot}}^{old}(s, \boldsymbol{a})\right] \; \geq \; \mathbb{E}_{\pi_{tot}^{old}}\left[Q_{\text{tot}}^{old}(s, \boldsymbol{a})\right].$$

Using the Bellman equation for the global Q-function:

$$Q_{tot}^{old}(s, \boldsymbol{a}) = R_{tot}(s, \boldsymbol{a}) + \gamma \mathbb{E}_{s', \boldsymbol{a}' \sim \pi_{tot}}\left[Q_{tot}^{old}(s', \boldsymbol{a}')\right],$$

where $R_{tot}$ is the cumulative reward of all agents. Then we can have:

$$
\begin{aligned}
Q_{tot}^{old} = \mathbb{E}_{s_1} & \left[R_{tot,t=0} + \gamma \mathbb{E}_{\mathbf{s_1} \sim \pi_{tot}^{old}}\left[Q_{tot}^{old}(s_1, \mathbf{a_1})\right]\right] \\
\leq & \mathbb{E}_{s_1}\left[R_{tot,t=0} + \gamma \left(\mathbb{E}_{\mathbf{a_1} \sim \pi_{tot}^{new}}\left[Q_{tot}^{old}(s_1, \mathbf{a_1})\right]\right)\right] \\
= & \mathbb{E}_{s_1}[R_{tot,t=0} + \gamma \left(R_{tot,t=1}\right) + \gamma^2 \mathbb{E}_{s_2}[\mathbb{E}_{\mathbf{a_2} \sim \pi_{tot}^{old}}\left[Q_{tot}^{old}(s_2, \mathbf{a_2})\right]]] \\
\leq & \mathbb{E}_{s_1}[R_{tot,t=0} + \gamma R_{tot,t=1}) + \gamma^2 \mathbb{E}_{s_2}\left[\mathbb{E}_{\mathbf{a_2} \sim \pi_{tot}^{new}}\left[Q_{tot}^{old}(s_2, \mathbf{a_2})\right]\right]] \\
= & \mathbb{E}_{s_1}\left[\alpha\gamma^\tau, s_2[R_{tot,t=0} + \gamma R_{tot,t=1}) + \gamma^2 R_{tot,t=2}\right. \\
& + \gamma^3 \mathbb{E}_{s_3}\left[\mathbb{E}_{\mathbf{a_3} \sim \pi_{tot}^{old}}\left[Q_{tot}^{old}(s_3, \mathbf{a_3})\right]\right]]] \\
& \vdots \\
\leq & \mathbb{E}_{\pi_{tot}^{new}}^{\infty}\left[\sum_{t=1}^{\infty} \gamma^t \left(R_{tot,t}\right)\right] \\
= & Q_{tot}^{new}(s, \boldsymbol{a}).
\end{aligned}
\tag{26}
$$

As each agent updates its individual state-action value function individually in turn, and since $Q_{\text{tot}}^{old}$ is monotonically non-decreasing under these local improvements, the procedure converges when *no agent can improve* $Q_{\text{tot}}$ further via changes in its individual state-action value which also means they attain the output from MRP. By construction, such a point must be globally optimal for $Q_{\text{tot}}$: otherwise, there would exist some local change in $\bar{Q}_i$ that continued to increase $Q_{\text{tot}}$, contradicting convergence. Hence, the final joint policy is also the global optimum for the original objective $Q_{\text{tot}}$, i.e.,

$$\boldsymbol{\pi}^* \; = \; \arg\max_{\pi_{tot}} \mathbb{E}_{\pi_{tot}}\left[Q_{\text{tot}}(s, \boldsymbol{a})\right].$$

$\square$

## B   Additional Experimental Results

### B.1   Fairness Analysis

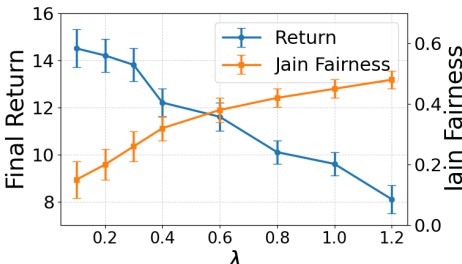

Figure 9: Trade-off between social welfare and fairness in the Escape Room environment ($N = 5$, $M = 3$) under different fairness weights $\lambda$.

Introducing a new mechanism or reshaping agents' utility functions inevitably alters the strategic structure of a non-cooperative game. This is a common approach in the literature on motivating cooperative behavior and resolving social dilemmas through incentive design (Itoh, 1991). This transformation raises an important question: *does reshaping incentives compromise fairness among agents?* In particular, some agents may consistently receive lower rewards under the original game, effectively bearing a disproportionate share of the social cost. Since in the real world, agents are not obliged to accept unfavorable outcomes, excessive unfairness may undermine participation and stability of the system.

To explicitly evaluate this issue, we assess the fairness impact of the proposed mechanism design framework using Jain's fairness index,

$$\mathcal{J}(R_1, \ldots, R_N) = \frac{(\sum_{i=1}^{N} R_i)^2}{N \sum_{i=1}^{N} R_i^2}, \tag{27}$$

This metric captures how evenly rewards are distributed across agents, with larger values indicating more equitable outcomes.

To control the trade-off between efficiency and fairness, we construct each agent's total training reward as

$$R_i^{\text{tot}} = \lambda R_i + \bar{R}_i,$$

where $\lambda \geq 0$ is a fairness weight. Intuitively, increasing $\lambda$ places greater emphasis on preserving agents' original payoffs.

Fig. 9 illustrates the resulting trade-off in the Escape Room environment with $N = 5$ and $M = 3$. As $\lambda$ increases, Jain's fairness index improves monotonically, indicating a more balanced distribution of returns across agents. At the same time, the overall social welfare decreases gradually, reflecting reduced specialization and a weaker push toward highly efficient but uneven equilibria. This behavior highlights a fundamental efficiency–fairness trade-off induced by mechanism-based utility reshaping.

## C   Hyper-parameters

In this section, we define the hyper-parameters and neural network structures used throughout the paper.

### C.1   Neural Network Structure

The Table 1 provides an overview of three neural network classes: counterfactual critic, maximized return predictor, and policy network. Each class has specific characteristics regarding its input, layer structure, activation function, and usage. The counterfactual critic takes a concatenation of the state shape and

| Class Name | Description |
| --- | --- |
| Counterfactual Critic Network | <ul><li>Input: Concatenation of state shape and actions</li><li>Layers:<ul><li>Linear layer (input: state shape + action dimensions, output: 128)</li><li>Linear layer (input: 128, output: 64)</li><li>Linear layer (input: 64, output: 32)</li><li>Output layer (input: 32, output: 1)</li><li>Dropout: 0.5</li></ul></li><li>Activation: ReLU</li></ul> |
| Maximized Return Predictor | <ul><li>Input: State shape</li><li>Layers:<ul><li>Linear layer (input: state shape, output: 64)</li><li>Linear layer (input: 64, output: 32)</li><li>Output layer (input: 32, output: 1)</li></ul></li><li>Activation: ReLU</li></ul> |
| Policy Network | <ul><li>Input: State size</li><li>Layers:<ul><li>Linear layer (input: state size, output: 64)</li><li>Linear layer (input: 64, output: 64)</li><li>Output layer (input:64, output: action size)</li></ul></li><li>Activation: ReLU</li></ul> |

Table 1: Neural Network Structures

actions as input and includes three linear layers with ReLU activations and dropout. It is used for Q-learning to predict the global state-action value and uses the Adam optimizer with a learning rate of 0.001. The maximized return predictor takes the state shape as input and has two hidden linear layers with ReLU activations, followed by an output layer that predicts the reward. It is used for value function approximation and also uses the Adam optimizer with a learning rate of 0.001. The policy takes the state size as input and includes two hidden linear layers with ReLU activations, followed by an output layer that predicts the action values. It is used for deep Q-learning and uses the Adam optimizer with a learning rate of 1e-4.

### C.2 Hyper-parameters

Experiments were conducted on hardware comprising an Intel(R) Xeon(R) Gold 6254 CPU @ 3.10GHz and four NVIDIA A5000 GPUs. This setup ensures the computational efficiency and precision required for the demanding simulations involved in multi-agent reinforcement learning and safety evaluations. Besides, Table 2 summarizes the default hyperparameters used across all experiments. We adopt a high discount factor ($\gamma = 0.99$) to emphasize long-term returns, which is crucial for coordination and safety-critical behaviors in multi-agent settings. Separate learning rates are assigned to the policy networks, counterfactual critic, and maximized return predictor to stabilize training under heterogeneous update dynamics. Gradient norm clip-

| Hyperparameter | Default Value | Description |
|---|---|---|
| $\gamma$ | 0.99 | Discount Factor |
| Individual Q-Learning Rate | 0.0001 | Policy nets learning rate |
| MRP Learning Rate | 0.001 | Learning rate of maximized return predictor |
| Critic Learning Rate | 0.001 | Learning rate of counterfactual critic |
| grad norm clip | 0.04 | clip rate of gradients |
| taus | 0.01 | Soft update weight |
| eps_decay | 5e-4 | Decay rate of epsilon |
| eps_min | 0.01 | Minimum epsilon |
| epsilon | 1 | Initial value for exploration |
| clip_min | 0.001 | Mimimum clip value for meta gradients |
| clip_max | 10 | Maximum clip value for meta gradients |

Table 2: Hyperparameters for Proposed Method

ping and bounded meta-gradient clipping are applied to mitigate exploding gradients and ensure numerical stability during meta-optimization. Soft target updates with a small $\tau$ further improve training smoothness.

## D  Algorithm Details

---
**Algorithm 1** Training Process
---
**Require:** Initialize environment, individual state-action value networks $\{\theta_i\}_{i \in \mathcal{N}}$, counterfactual critic parameters $\psi$, maximized return predictor parameters $\phi$, and replay buffers $\mathcal{B}$ and MRP buffer $\mathcal{D}$.
 1: **for** episode $= 1$ to $M$ **do**
 2:    Reset the environment and receive the initial state $s_0$.
 3:    **for** $t = 1, \ldots, H$ **do**
 4:       Observe the current state $s^t$.
 5:       **for** $i \in \mathcal{N}$ **do**
 6:          Select action $a_i^t \sim \pi_{\theta_i}(\cdot \mid s^t)$ using $\epsilon$-greedy strategy.
 7:          Execute action $a_i^t$, receive reward $R_i^t$, and observe new state $s^{t+1}$.
 8:          Store transition tuple $(s^t, \boldsymbol{a}^t, R^t, s^{t+1})$ in replay buffer $\mathcal{B}$.
 9:       **end for**
10:       Sample a mini-batch of transitions $\{(s^k, \boldsymbol{a}^k, R^k, s^{k+1})\}_{k=1}^{B}$ from $\mathcal{B}$.
11:       Update MRP buffer $\mathcal{D}$ using the states and their corresponding joint state-action values by Eq. 14.
12:       Update counterfactual critic by minimizing loss using Eq. 13.
13:       Update maximized return predictor by minimizing loss using Eq. 16.
14:       Compute mechanism reward $\bar{R}$ using Eq. 9.
15:       **for** $i \in \mathcal{N}$ **do**
16:          Update individual state-action value network $\theta_i$ using value iteration Eq. 24.
17:       **end for**
18:    **end for**
19: **end for**
---

